# Chimeric Peptidoglycan Hydrolases Kill Staphylococcal Mastitis Isolates in Raw Milk and within Bovine Mammary Gland Epithelial Cells

**DOI:** 10.3390/v14122801

**Published:** 2022-12-15

**Authors:** Anja P. Keller, Shera Ly, Steven Daetwyler, Fritz Eichenseher, Martin J. Loessner, Mathias Schmelcher

**Affiliations:** Institute of Food, Nutrition and Health, ETH Zurich, 8092 Zurich, Switzerland

**Keywords:** *Staphylococcus aureus*, MRSA, mastitis, endolysins, peptidoglycan hydrolases, cell-penetrating peptide

## Abstract

*Staphylococcus aureus* is a major causative agent of bovine mastitis, a disease considered one of the most economically devastating in the dairy sector. Considering the increasing prevalence of antibiotic-resistant strains, novel therapeutic approaches efficiently targeting extra- and intracellular bacteria and featuring high activity in the presence of raw milk components are needed. Here, we have screened a library of eighty peptidoglycan hydrolases (PGHs) for high activity against *S. aureus* in raw bovine milk, twelve of which were selected for further characterization and comparison in time-kill assays. The bacteriocins lysostaphin and ALE-1, and the chimeric PGH M23LST(L)_SH3b2638 reduced bacterial numbers in raw milk to the detection limit within 10 min. Three CHAP-based PGHs (CHAPGH15_SH3bAle1, CHAPK_SH3bLST_H, CHAPH5_LST_H) showed gradually improving activity with increasing dilution of the raw milk. Furthermore, we demonstrated synergistic activity of CHAPGH15_SH3bAle1 and LST when used in combination. Finally, modification of four PGHs (LST, M23LST(L)_SH3b2638, CHAPK_SH3bLST, CHAPGH15_SH3bAle1) with the cell-penetrating peptide TAT significantly enhanced the eradication of intracellular *S. aureus* in bovine mammary alveolar cells compared to the unmodified parentals in a concentration-dependent manner.

## 1. Introduction

Bovine mastitis is considered one of the most economically devastating diseases in the dairy sector. The estimated financial loss per cow of 131 USD each year (corresponding to ca. 15% of gross profit) is mainly due to decreased milk production, poor milk quality and the premature culling of affected animals to prevent further events [1,2]. Infections can spread to other animals in a herd by way of close contact, shared milking equipment or transmission by the milker [3]. The disease presents as an inflammation of the mammary glands and is most frequently caused by bacterial intra-mammary infections [4]. Depending on the origin of the infectious agent, contagious and environmental mastitis are differentiated [5]. Up to 40% of lactating cows experience bovine mastitis at least once in their lifetime, and it is estimated that on a single farm, up to 30% of lactating cows present with subclinical mastitis at any given time [2]. Although the prevalence of mastitis has improved over the last decades in countries with a well-established dairy industry, it is still widespread in developing countries [6]. Several recent studies on the prevalence of mastitis reported high percentages of affected animals in Africa. In Ethiopia, a prevalence of 62.6% infected cows was reported (*n* = 529, randomly selected from 95 herds), and *S. aureus* was isolated from 73.2% of herds affected with mastitis [7]. Similar numbers were observed for sub-clinical cases of mastitis in Uganda (68.6%, *n* = 608 cows from 30 farms [8]) and Rwanda (62%, *n* = 572 from 404 herds [9]).

*Staphylococcus aureus* is a major causative agent of both clinical and sub-clinical bovine mastitis, which can progress into long-lasting chronic infections [10,11]. Sub-clinical mastitis is often not detected due to a lack of symptoms, which can be harmful to consumers due to the presence of *S. aureus* or its enterotoxins in milk products [12,13,14,15]. Treatment of affected cows is commonly done after milking by intra-mammary infusion of broad-spectrum antibiotics such as penicillins, pirlimycin and advanced cephalosporins [16,17]. The increasing frequency of antibiotic resistance, together with the ability of *S. aureus* to invade mammary gland cells and to form biofilms, render the treatment of staphylococcal mastitis difficult [18]. Novel antimicrobials are needed that fulfil the requirements of efficacy against intracellular persisters, biofilms and resistant strains, while not contributing to further spread of antimicrobial resistance themselves.

In the last decade, several studies have highlighted the potential of peptidoglycan hydrolases (PGHs), such as bacteriocins and bacteriophage-derived endolysins, for the treatment of mastitis (extensively reviewed in [19]). PGHs efficiently and rapidly lyse bacterial cells by cleaving specific bonds in the peptidoglycan, an essential structural element of the bacterial cell wall. This occurs independently of an active bacterial metabolism, and PGHs have indeed been shown to be effective against dormant persister cells in biofilms and infected host cells, which are central scenarios in the treatment of mastitis [20,21,22]. In endolysins targeting Gram-positive bacteria, enzymatic activity and high-affinity recognition of cell wall target structures are spatially separated into distinct domains: enzymatically active domains (EAD) and cell-wall binding domains (CBD) [23]. This modularity is also observed for some bacteriocins, such as lysostaphin (LST) and its homologue ALE-1 [24,25], and has been exploited in the production of chimeric enzymes with potentiated activity against *S. aureus* [26]. Based on their enzymatic activity, PGHs can be grouped into muramidases, glucosaminidases, amidases and endopeptidases [27]. Combined applications of PGHs with activity against different structural targets have been shown to kill *S. aureus* in a synergistic fashion [21,22,28,29,30]. Further, endolysins are effective against antibiotic-resistant strains [27,31] and display high target specificity on a genus level (some on a serovar or strain level), thus significantly reducing off-target effects on the healthy microbiome [32]. These characteristics highlight the potential of chimeric PGHs and endolysins as antimicrobial agents to combat staphylococcal mastitis.

Previous studies have recognized the complexity of raw bovine milk as a medium, as its presence can drastically affect the in vitro and in vivo activity of PGHs [21,30]. As a treatment regime would likely involve intra-mammary infusion of the enzymes, a therapeutic activity in the presence of milk residues is of utter importance. Verbree et al. (2018) have developed a method to rapidly screen large libraries of PGHs to identify enzymes retaining high staphylolytic activity in processed milk [30]. In their study, 8 of over 170 screened constructs showed promising activity, and the 2 enzymes with the highest activity (LST and the chimeric CHAPK_SH3bLST) presented synergistic activity in processed milk. However, only LST retained high activity in raw milk.

As endolysins have low inherent capabilities to cross eukaryotic membranes, targeting of intracellularly residing persisters poses a further challenge in the eradication of sub-clinical and chronic staphylococcal mastitis. Cell-penetrating peptides (CPP), i.e., small peptides capable of crossing mammalian membranes and mediating intracellular translocation of associated cargo molecules, have been proposed as a solution and recently successfully applied in the treatment of intracellular bacteria in a murine abscess model [22,27,33]. This approach was recently also implemented by Becker et al., who demonstrated targeting of intracellular *S. aureus* in bovine mammary epithelial cells and a murine model of mastitis using chimeric PGHs modified with CPPs [21].

Here, we expanded on the previous works by screening for novel chimeric PGHs with high activity directly in raw bovine milk. We characterized the activity of twelve selected constructs in undiluted and diluted raw milk in time-kill assays and determined the minimum bactericidal concentrations (MBC) of seven PGHs in raw and processed bovine milk and against several mastitis isolates and a methicillin-resistant *S. aureus* (MRSA) strain. Further, we assessed the synergistic activity of two endopeptidases of the CHAP family (CHAPK_SH3bLST and CHAPGH15_SH3bAle1) with LST and its chimeric derivative M23LST(L)_SH3b2638. Finally, we demonstrated the concentration-dependent killing of intracellular *S. aureus* in bovine mammary alveolar cells by four PGHs modified with a cell-penetrating peptide.

## 2. Materials and Methods

### 2.1. Bacterial Strains and Growth Conditions

*Escherichia coli* and *Staphylococcus aureus* strains used in this study are listed in Table 1 and Appendix A. *S. aureus* were grown in Tryptic Soy Broth (TSB) and *E. coli* were grown in Luria-Bertani medium (LB, 10 g/L peptone, 5 g/L yeast extract, 5 g/L NaCl) or LB-PE optimized for protein expression (15 g/L peptone, 8 g/L yeast extract, 5 g/L NaCl) [34]. Bacterial cultures were supplemented with appropriate antibiotics, as listed in Table 1 and Appendix A. *E. coli* expression strains were additionally supplemented with antibiotics to select for expression plasmids: ampicillin (100 µg/mL) for pET302 (Thermo Fisher Scientific, Waltham, MA, USA), pQE30 (Qiagen, Hilden, Germany) and pET21a (Merck, Darmstadt, Germany), or kanamycin (50 µg/mL) for pET9a (Merck, Darmstadt, Germany) and pET200 (Thermo Fisher Scientific, Waltham, MA, USA). All bacteria were cultured at 37 °C and liquid cultures were grown shaking at 180 rpm.

### 2.2. Screening of PGH Constructs for High Staphylolytic Activity in Raw Bovine Milk

A total of 80 enzymes with novel or improved domain pairings were selected from our in-house collection of over 380 peptidoglycan hydrolases. Enzymes were screened for high activity in raw bovine milk in a microwell plate setting essentially as described previously [30]. In short, *E. coli* strains were inoculated in a 96-well plate and proteins were expressed as described. Cells were lysed by exposure to chloroform vapor. Fresh bovine raw milk (Strickhof, Lindau, Germany) was spiked with *S. aureus* Newbould 305 (10^5^ CFU/mL) and added to each well containing exposed PGHs. The plate was incubated at 37 °C for two hours. Contents of wells were diluted tenfold in DPBS (Thermo Fisher Scientific, Waltham, MA, USA), and 7.5 µL were spotted on Baird Parker agar plates supplemented with Egg Yolk Tellurite Emulsion 20% (BP-EYT, both from Biolife Italiana, Milan, Italy) according to the manufacturer’s instructions and incubated at 37 °C overnight. *E. coli* strains carrying vectors pET21a and pET21a_LST were used as negative and positive controls, respectively. Technical replicates were evaluated and scored according to the following scheme: 0 colonies, 3; ≤10 colonies, 2; ≤20 colonies, 1; ≥20 colonies, 0. Scores from all technical and biological replicates were added, resulting in a maximum possible score of 27. Scores reached by the 12 PGHs selected for further experiments are represented in Table 1 as follows: 0 to 14, −; 15 to 19, +; 20 to 24, ++; 25 to 27, +++.

### 2.3. Expression and Purification of Proteins

PGH constructs were expressed as described previously [22,30]. Briefly, *E. coli* strains harbouring plasmids containing constructs of interest were grown to mid-log phase in LB-PE, cooled on ice and induced with 0.5 mM IPTG (Carl Roth GmbH, Karlsruhe, Germany). Cultures were incubated for 18 h at 19 °C. Cells were harvested by centrifugation (7000× *g*, 10 min, 4 °C; Avanti^®^ J-E Centrifuge, JA-10 rotor; Beckman Coulter, Brea, CA, USA) and pellets were frozen at −20 °C. Pellets were resuspended in CIEX Lysis Buffer (50 mM Na_2_HPO_4_, 20% glycerol, pH = 7.4) or IMAC Lysis Buffer (300 mM NaCl, 50 mM Na_2_HPO_4_, 10 mM imidazole, 30% glycerol, pH = 8) for constructs without and with His-tags, respectively, and lysed by pressure homogenization (100 MPa; Stansted Pressure Cell Homogeniser; Homogenising Systems Ltd., Harlow, UK). After removal of cellular debris by centrifugation (20,000× *g*, 60 min, 4 °C; Sigma 3K30, rotor 19776; Sigma-Aldrich, St. Louis, MO, USA), proteins were purified from cleared lysates. Proteins without a histidine tag were purified by cation exchange chromatography (CIEX) on an Äkta pure FPLC system (GE Healthcare, Chicago, IL, USA) with a 5 mL HiTrap SP FF column (Cytiva, Marlborough, MA, USA). Elution of target proteins was executed in CIEX Lysis Buffer over an increasing sodium chloride gradient. Proteins with a 6x histidine tag were purified on a gravity flow column packed with a nickel-nitrilotriacetic acid (Ni-NTA) resin (ABT, Madrid, Spain). Briefly, the protein-containing supernatant was incubated for one hour at 4 °C to allow binding of the protein to the resin. Resin was washed with 30 column volumes IMAC Lysis Buffer and proteins were eluted with Elution Buffer (IMAC Lysis Buffer containing 250 mM imidazole). Protein concentrations were determined by NanoDrop measurements (Nanodrop^®^ ND-1000; Thermo Fisher Scientific, Waltham, MA, USA). Size and purity of proteins were verified by SDS-PAGE on Criterion™ TGX Stain-Free™ precast gels (Bio-Rad, Hercules, CA, USA) and gels were stained with Coomassie (InstantBlue™, Abcam, Cambridge, UK).

### 2.4. Assessment of Staphylolytic Activity of PGHs

In vitro activity of PGHs was evaluated in time-kill assays, and the minimum bactericidal concentration (MBC) was determined as previously described [30]. **Time-kill assay**: In short, enzymes were diluted in TSB medium, raw or UHT milk to a 10× concentrated stock (to achieve final concentrations in the assay of 20 and 100 nM for TSB, 360 nM for diluted raw milk, undiluted raw milk and UHT milk) and transferred to a 96-well plate (20 µL per well). An overnight culture of *S. aureus* Newbould 305 was grown to mid-log growth phase in fresh TSB medium and diluted to 1.1 × 10^6^ CFU/mL in growth medium or milk. 180 µL were added to the enzymes and the plate was incubated at 37 °C. Surviving bacteria and inoculum were enumerated after 0, 10, 60 and 180 min from a dilution series and spot plating (LB or BP-EYT agar). Plates were incubated overnight at 37 °C and colonies were counted the next day. The number of surviving bacteria in CFU/mL was plotted against the time in minutes, and log reductions compared to the untreated control were calculated for the 180-min timepoint. **Minimum bactericidal concentration**: Due to the turbidity of raw milk, a classical minimum inhibitory concentration assay based on the determination of bacterial growth by optical density measurements cannot be applied. As an alternative, the minimum bactericidal concentration (MBC) assay was chosen, where the lowest concentration needed to kill a target organism is determined by plating. PGHs were tested against multiple *S. aureus* mastitis isolates in UHT milk, raw milk and raw milk diluted in DPBS to 30% and 10%. All assays were performed in biological triplicates. Bacterial overnight cultures were grown to mid-log phase and diluted to 5 × 10^5^ CFU/mL in milk or growth medium. Enzymes were serially diluted in the corresponding condition at 2× the final concentration. Bacteria and enzymes were mixed 1:1 and incubated at 37 °C for 120 min, shaking at 200 rpm. 7.5 µL of each dilution were spotted in duplicates on square BP-EYT agar plates and incubated for 18 to 20 h at 37 °C. The MBC value was determined from the spot with the lowest concentration where clearance up to a threshold of four colonies was observed.

### 2.5. Cell Culture

Bovine mammary alveolar cells (Mac-T; provided by Olga Wellnitz at Agroscope, Posieux, CH) were cultured in DMEM (1X) + GlutaMax AX™-I ([+] 4.5 g/L D-Glucose, [+] 110 mg/L Sodium Pyruvate; Thermo Fisher Scientific, Waltham, MA, USA) supplemented with 10% FBS (Gibco^®^, certified, OneShot™; Thermo Fisher Scientific, Waltham, MA, USA) and 4 mM L-glutamine (Carl Roth GmbH, Karlsruhe, Germany). Cells were grown in T75 flasks (VWR, Radnor, PA, USA). For maintenance and all experimental incubation steps, cells were grown at 37 °C in a 5% CO_2_ atmosphere and split every three to four days at a ratio of 1:10.

### 2.6. Intracellular Activity Assay

To assess the intracellular staphylolytic activity of selected PGHs with high activity in intramammary conditions, an intracellular killing assay was performed essentially as previously described [22]. The cell-penetrating peptide TAT [35] was selected to mediate transfer of the enzymes into eukaryotic cells. Fusion proteins consisting of PGHs and C-terminal TAT peptides (PGH_TAT) were generated as previously described [22]. The in vitro activity of freshly expressed batches of parental PGHs and PGH_TAT variants was first determined in a time-kill assay in different raw milk conditions. For the intracellular killing assay, Mac-T cells were seeded in a 24-well plate (1 × 10^5^ cells/well in 1 mL) and grown for 24 h. Eukaryotic cells were washed once with culture medium and infected with *S. aureus* Newbould 305 at an MOI of 50. As controls, one well was not infected and one infected but not treated with any enzyme. The plate was subjected to a quick centrifugation step (130× *g* for 5 min; Sigma 2–16 K, rotor 11123; Sigma-Aldrich, St. Louis, MO, USA), and eukaryotic cell invasion was carried out for one hour. To eliminate extracellular bacteria, cells were washed once with DPBS and incubated for one hour in culture medium supplemented with flucloxacillin (1 mg/mL; Acros Organics, Geel, Belgium). Two µM of PGH or PGH_TAT in culture medium supplemented with flucloxacillin were incubated with the cells for three hours. After three DBPS washing steps, cells were detached and lysed using Trypsin and a 0.1% TritonTM X-100 solution (both Thermo Fisher Scientific, Waltham, MA, USA). Additionally, each well was manually resuspended 40 times. DPBS was added to each lysate to reach a final volume of 1 mL. The 1:10 dilutions were spotted onto LB agar plates in triplicates, and plates were incubated overnight at 37 °C. Colonies were counted the next day.

### 2.7. Synergistic Activity of PGHs in Milk

To show synergistic killing of bacteria in UHT milk, PGHs with different enzymatic activity were paired and compared to the activity of parental enzymes on their own. A time-kill assay was performed as described above. Bacterial numbers were only determined for the inoculum and a final timepoint of 60 min. Concentrations of parental enzymes were chosen so that the log (CFU/mL) reduction at the endpoint compared to the control was in a similar range. To show synergy, the two parental enzymes were mixed at half the pre-determined concentration, and the activity of the mixture was compared to the activity of the single parental enzymes. Synergy presents as a significantly higher log(CFU/mL) reduction for the combination than for the single PGHs.

### 2.8. Statistical Analysis

All statistical analyses shown were performed in GraphPad Prism 9.2.0 (GraphPad Software, San Diego, CA, USA). Analysis of the intracellular killing assay was performed using a Kruskal–Wallis test and subsequent Dunn’s multiple comparison, comparing the TAT-modified variants to their parental PGH control. Data from the synergy experiments were log transformed (log (CFU/mL)) and analyzed using a Friedman test. This was followed by Dunn’s multiple comparison to compare the mixture of both PGHs to the single PGH with the higher activity. All experiments were performed in at least biological triplicates. *P*-values < 0.05 were deemed significant, and p-values are represented in the figures as follows: ns, *p* > 0.05; *, *p* ≤ 0.05; **, *p* ≤ 0.01; ***, *p* ≤ 0.001; ****, *p* < 0.0001.

## 3. Results

### 3.1. Identification of PGHs with High Staphylolytic Activity in Raw Milk

Over 170 constructs from our in-house collection had previously been screened for high activity in UHT milk [30]. Since then, the collection has been considerably expanded and now holds over 380 parental and chimeric constructs. In an effort to identify novel therapeutics against bovine mastitis, we screened eighty enzymes with novel or improved domain architectures for high staphylolytic activity using the previously established microwell plate method [30]. The screening was performed directly in raw bovine milk, a highly complex environment that exerts a stricter selective pressure compared to UHT milk and more closely mimics the conditions encountered in an applicational setting. A total of 12 PGHs, including the positive control lysostaphin (LST), showed high activity in raw milk (Table 1), reaching scores equal to or higher than 15 (out of 27). Besides LST, 8 constructs reached the maximum possible score of 27, which corresponds to a consistent and complete eradication of bacteria in this experimental setting. The domains represented in this selection originate from a broad range of native peptidoglycan hydrolases, including lysostaphin (LST) [36] and ALE1 [25], two homologous bacteriocins originating from *Staphylococcus simulans* and *Staphylococcus capitis*, respectively. The remaining domains stem from endolysins of the bacteriophages 2638A [37], GH15 [38], K [39], H5 [40] and Twort (Tw) [41]. Due to the diverse backgrounds of the screened constructs, the 12 best PGHs are comprised of various types of enzymatically active domains, including Gly-Gly endopeptidases (M23 domains; PGHs 1–6), Gly-d-Ala endopeptidases (CHAP domains; PGHs 7, 8) or combinations thereof (PGHs 9–12). In a next step, the activity of the selected PGHs was quantified for further distinction.

**Table 1 viruses-14-02801-t001:** Top scoring PGHs screened for high staphylolytic activity in raw milk. PGHs are sorted according to their enzymatic activity.

ID	Name	Expression Vector ^1^	Expression Strain (*E. coli*)	Activity in Screening ^2^
*Gly-Gly Endopeptidases*				
PGH 1	LST	pET302	BL21-Gold(DE3)	+++
PGH 2	M23LST(L)_SH3b2638	pET302	BL21-Gold(DE3)	+++
PGH 3	ALE-1	pET302	BL21-Gold(DE3)	+++
PGH 4	H_LST_M23LST	pQE30	SURE	++
PGH 5	H_TEV_(M23LST)2_SH3bLST	pQE30	SURE	++
PGH 6	(M23LST)2_SH3bAle1	pET302	BL21-Gold(DE3)	+
*Gly-* d *-Ala Endopeptidases*				
PGH 7	CHAPGH15_SH3bAle1	pET302	BL21-Gold(DE3)	++
PGH 8	CHAPK_SH3bLST_H	pET21a	BL21-Gold(DE3)	++
*Double EAD mixed specificity*				
PGH 9	CHAPH5_LST_H	pET21a	BL21-Gold(DE3)	+++
*Triple EAD mixed specificity*				
PGH 10	LST_CHAPK_AmiK_H	pET21a	BL21-Gold(DE3)	+++
PGH 11	LST_CHAPH5_AmiH5(L)_H	pET21a	BL21-Gold(DE3)	+++
PGH 12	H_CHAPTw_Ami2638_M23LST_SH3b2638	pQE30	XL1-Blue MRF’	+

^1^ All plasmids carry an ampicillin resistance gene. ^2^ Translation of scores from screening to activity: 15–19, +; 20–25, ++; 26–27, +++.

### 3.2. Selected PGHs Show High Activity against Staphylococcal Bovine Mastitis Isolates

After expression and purification of the 12 selected enzymes with scores ≥ 15, we determined their activity against the mastitis isolate *S. aureus* Newbould 305 in standard TSB growth medium (20 nM and 100 nM) and raw milk (360 nM). The latter was applied undiluted and diluted in DPBS to mimic the environment of an intramammary treatment setting [30]. To compare their staphylolytic activity over time, equimolar concentrations of the 12 PGHs were mixed with bacteria and incubated for 180 min. The inoculum was set at 10^6^ CFU/mL, which is similar to the bacterial load observed in acute-stage bovine mammary gland infections [30]. Surviving bacteria were enumerated by plating after 0, 10, 60 and 180 min (Appendix A). After 180 min, the log reduction in CFU/mL in comparison to an untreated control was determined for each tested condition (Figure 1). We found that all enzymes reduced the bacterial load in TSB by at least 1.4 log units (at 100 nM), except for H_CHAPTw_Ami2638_M23LST_SH3b2638. Generally, larger constructs with four domains showed lower activity in the tested conditions as compared to smaller enzymes with fewer domains. Notably, LST, ALE1 and M23LST(L)_SH3b2638 reduced bacterial numbers to undetectable levels in all conditions after 10 min, including undiluted raw milk. For other constructs, the performance improved with increased dilution of raw milk. This was especially pronounced for CHAPK_SH3bLST, CHAPGH15_SH3bAle1 and CHAPH5_LST_H, where a 10-fold higher reduction in bacterial numbers was observed in 10% raw milk as compared to undiluted raw milk.

Based on these results, we excluded five constructs from further experiments, as they showed inadequate activity. The remaining seven constructs were tested against five mastitis isolates and the MRSA strain USA300 in UHT milk to assess if high staphylolytic activity was transferrable to other staphylococcal strains (Figure 2, Appendix A). The minimum bactericidal concentration (MBC) was assessed in TSB, raw milk and diluted raw milk. The obtained MBCs correlated with the activity observed in time-kill assays, as enzymes containing an M23 domain exhibit lower MBCs than CHAP-based enzymes. Lysostaphin and the homologous ALE-1 showed the lowest MBC against all tested strains, followed closely by the chimeric M23LST(L)_SH3b2638. Generally, the average MBC of an enzyme showed minor differences depending on the targeted strain (e.g., LST: 23.4 ± 4.5 nM (Newbould, *n* = 3) vs 7.8 ± 0.0 nM (M2071, *n* = 3) vs. 11.7 ± 2.6 nM (USA300, *n* = 3)).

To further investigate the effect of raw milk on the enzymatic activity, MBCs against *S. aureus* Newbould 305 were determined in undiluted and progressively more diluted raw milk (Figure 3, Appendix A). In line with our previous results, MBCs decreased with increasing dilution of the raw milk for all enzymes, demonstrating the inhibitory effect raw milk components have on the enzymatic activity of PGHs. Nevertheless, the initial screening assay allowed for the selection of several enzymes with suitable activity for further evaluation.

### 3.3. Lysostaphin Shows Synergistic Activity with Two CHAP-Based PGHs

Synergistic activity of PGHs with differing enzymatic activity has previously been demonstrated in milk [30] and a murine model of staphylococcal mastitis [42]. Verbree et al. demonstrated synergy for a combination of LST and CHAPK_SH3bLST [30]. Here, we assessed if this is a general feat of combinations of enzymes with M23 and CHAP endopeptidase function. To this end, we combined LST or the chimeric M23LST(L)_SH3b2638 (M23) with either of the two best performing CHAP endopeptidases: CHAPK_SH3bLST (CHAPK) and CHAPGH15_SH3bAle1 (CHAPGH15). Synergy was determined in UHT milk, based on the time-kill assay method against *S. aureus* Newbould 305 (10^6^ CFU/mL). First, the concentrations of the individual enzymes were adjusted so that they yielded a similar log reduction in bacterial numbers after one hour (10 nM for LST and M23, 20 nM for CHAPGH15 and 25 nM for CHAPK). To demonstrate synergy, two PGHs were combined at half the concentration of the single treatments (Figure 4, e.g., 5 nM LST + 10 nM CHAPGH15). LST showed synergistic killing with both CHAPK and CHAPGH15, confirming the results of Verbree et al. In addition, our data highlights CHAPGH15_SH3bAle1 as a strong candidate for a combination treatment approach. Interestingly, M23LST(L)_SH3b2638 did not show synergistic effects, although the enzymatically active domain of this construct is identical to the one in LST, with the two enzymes differing only in their C-terminal cell wall binding domain. Overall, these results corroborate the potential of applying synergistic cocktails of several PGHs for the treatment of staphylococcal infections.

### 3.4. Modification with Cell-Penetrating Peptides Enhances the Efficacy of PGHs against Intracellular Bacteria

*S. aureus* has been shown to infect mammary gland epithelial cells [43]. As such, intracellular persisters are associated with recurrent infections, and targeting intracellular bacteria should be considered critical in the development of an effective treatment of staphylococcal mastitis. One approach is the modification of PGHs with a cell-penetrating peptide, which mediates uptake of its cargo into eukaryotic cells [21,22,44]. Here, we modified the PGHs LST, M23LST(L)_SH3b2638, CHAPGH15_SH3bAle1 and CHAPK_SH3bLST with the well-characterized cell-penetrating peptide TAT [45]. The activity of all parental enzymes in raw milk was significantly reduced upon C-terminal addition of TAT (Appendix A). While the same pattern could be observed for the CHAP-based constructs in 10% raw milk, M23LST(L)_SH3b2638, LST and their TAT-modified variants reduced bacterial numbers to the detection limit within 10 min under these conditions. TAT-modified PGHs and parental PGHs were used to treat bovine mammary alveolar cells infected with *S. aureus* Newbould 305 (Figure 5). After infection and lysis, untreated MC3T3-E1 cells displayed a stable intracellular infection of roughly 10^6^ CFU/mL, which corresponds to roughly 5 CFU/cell. Parental enzymes only slightly affected these levels of intracellular bacteria, whereas modification with the cell-penetrating peptide TAT significantly improved intracellular killing (LST_TAT, *p* = 0.0005; M23_TAT, *p* = 0.0003; CHAPK_TAT, *p* = 0.015; CHAPGH15_TAT, *p* = 0.002). This effect was most pronounced for LST_TAT, which showed a significant reduction (*p* = 0.027) even at one fourth of the concentration of the parental enzyme. 

## 4. Discussion

PGHs have been studied as novel antimicrobial agents for the treatment of a variety of staphylococcal diseases [46]. The optimization of PGHs by protein engineering is facilitated by their modular structure, and many chimeric PGHs have proven to be more effective than their naturally occurring parentals [21,30,47,48,49,50]. The different physiological conditions encountered by PGHs in specific treatment scenarios can markedly affect their activity, and the selection of constructs with optimal activity for a specific application has been deemed an important step in the advancement of PGH therapeutics research. Thus, several strategies, which allow the efficient screening of libraries of chimeric PGHs to identify constructs with increased activity and stability under relevant conditions, have been developed [30,51,52]. These methods have been employed to identify PGHs with high antimicrobial activity in murine and human serum [48,50,52,53], increased intracellular efficacy [22] and extended host range [51]. Previously, Verbree and colleagues developed a microwell plate-based screening method to identify enzymes with high staphylolytic activity in bovine UHT milk for the treatment of bovine mastitis from our in-house library of natural, truncated and chimeric PGHs [30]. As this library has been significantly expanded in the last years, we have now applied this screening method to 80 new PGHs. The expanded library contains natural PGHs, novel architectures and previously designed constructs that have been optimized since. 

Milk is a complex environment for the application of antimicrobials due to the presence of milk fat globules and whey proteins that interact with *S. aureus*, shielding the bacterium from antimicrobial agents [54,55,56]. Similarly, heat-sensitive agglutinating milk components have been reported to mediate the formation of bacterial cell clumps and the clustering of fat globules, which might limit the access of PGHs to their bacterial targets [57,58]. Moreover, milk contains a variety of proteases, which could inactivate the applied PGHs [59]. These effects are alleviated by heat-treatment of the raw milk. Leite et al. have shown a significant decrease in the activity of non-specific proteases in bovine raw milk upon heat treatment, which was more pronounced for ultra-high temperature-treated milk than pasteurized milk [60]. Only a few bacteriophage-derived PGHs have been reported to show activity in bovine milk, and many were observed to be more active in heat-treated milk than raw milk, and more active in skim milk than whole milk [30,40,42,47,61,62,63,64,65]. Acknowledging the stricter selective pressure of whole raw milk over heat-treated milk, we have performed our screening directly in raw milk. Twelve PGHs showed promising staphylolytic activity, nine of them consistently facilitating complete eradication of bacteria (Table 1). Several PGHs in this selection had also been identified in the previous screening in UHT milk by Verbree et al., namely LST, CHAPK_SH3bLST_H and CHAPH5_LST_H. We have included LST as a positive control, as it had previously demonstrated high activity in raw milk [30]. The two constructs CHAPK_SH3bLST_H and CHAPH5_LST_H were expressed from an optimized system, which involves a new expression host (*E. coli* BL21-GoldDE3) and plasmid (pET21a). Further, our selection included the chimeric PGH M23LST(L)_SH3b2638, which contains an adjusted linker region, that confers improved staphylolytic activity in a variety of conditions compared to its predecessor M23LST_SH3b2638 (data not shown). The fact that these PGHs were identified in raw and UHT milk demonstrates the robustness and validity of the screening method to select enzymes with high activity under given conditions.

We have further characterized the selected constructs by quantifying their lytic activity over time in raw milk (Figure 1). LST reduced the bacterial numbers to the detection limit within ten minutes, confirming the findings by Verbree et al. The same was observed for M23LST(L)_SH3b2638 and Ale1, which is not surprising, as the former is a chimeric derivative (albeit with a bacteriophage endolysin-derived C-terminal CBD) and the latter a close homologue of LST [25]. The three enzymes also consistently demonstrated the lowest MBCs against *S. aureus* mastitis isolates in UHT and raw milk (Figure 2 and Figure 3). The performance of the three enzymes, CHAPK_SH3bLST, CHAPGH15_SH3bAle1 and CHAPH5_LST_H, strongly correlated with increasing dilution of the raw milk with DPBS, which might be due to the reduction in milk fat globule density and its inhibitory effect. The inhibitory effect of raw milk components was evident in the time-kill assay data (Figure 1) and from the determined MBC values against *S. aureus* Newbould 305 (Figure 3). Interestingly, a recent study analysed the effect of milk components on the activity of CHAPGH15 and the full-length parental LysGH15 [66]. Although the full-length endolysins were strongly affected by the presence of milk fat (>95% activity loss), they observed only a small impact on the activity of the single EAD CHAPGH15 (10% activity loss). Together with our findings, this seems to imply that the addition of any C-terminal domain inhibits the activity of the CHAPGH15 domain in raw whole milk, although not in other media or diluted milk, and the mechanisms behind this will have to be further explored. While CHAPGH15_SH3bAle1 was not active in whole raw milk, it retained high activity in diluted raw milk, which mimics the conditions encountered in an intramammary treatment setting, where residual milk components mix with the carrier solution (likely a buffer) of the PGH therapeutic [30]. Further, we have observed the impact of heat-treatment of the milk on the activity of PGHs. Where the enzymes containing an M23 endopeptidase domain showed similar MBCs in raw milk and heat-treated milk, the two CHAP domain-containing enzymes CHAPK_SH3bLST and CHAPGH15_SH3bAle1 showed 12-fold and 4.5-fold higher MBCs in raw milk, respectively. A similar effect was observed by Rodriguez-Rubio and colleagues, who found the staphylolytic activity of a chimeric PGH (CHAPSH3b: N-terminal CHAP domain from virion-associated PGH HydH5 and C-terminal SH3b domain of LST) to be higher in heat-treated milk than raw milk [62]. They have also observed the inhibitory effect of milk fat, as CHAPSH3b was more active in heat-treated skim milk than heat-treated whole milk, which is in accordance with our findings (Figure 2 and Figure 3).

Despite its superior activity as compared to most PGHs described to date, it has been suggested that LST should be used only in combination with other antimicrobials, as it is able to induce resistance in *S. aureus* in experimental settings, although to this day only a few natural LST-resistant isolates have been found [67,68,69,70,71]. Lysostaphin cleaves the pentaglycine crossbridge characteristic for the *S. aureus* peptidoglycan, and alterations therein have been attributed to mutations in the glycine transferase-encoding genes *femA* and *femB* (shortening of interpeptide bridge) or genes encoding serine transferases, such as *lif*, *glyA* or *shmT* (replacement of glycine by serine) [71,72,73]. It has been proposed that the combined application of PGHs with different cleavage sites reduces the risk of resistance formation, as it is very unlikely for a bacterium to carry two life-saving mutations at once [74]. Interestingly, LST was shown to act synergistically with a variety of antimicrobials, including antibiotics [75,76,77], antimicrobial peptides [78,79] and endolysins [22,28,30,63]. Synergy was observed between LST and the phage K-derived endolysin LysK [28], and later was also demonstrated with the chimeric CHAPK_SH3bLST in milk against several mastitis isolates [30]. We have assessed the synergistic activity of CHAPGH15 with LST and its chimeric derivative M23LST(L)_SH3b2638 in milk. CHAPGH15_SH3bAle1 showed similarly strong synergy with LST as CHAPK_SH3bLST. Interestingly, the two enzymes did not show any synergy with M23LST(L)_SH3b2638, demonstrating that the synergy between two PGHs is not solely dependent on the presence of two enzymatic domains with different cell wall targets, as LST and M23LST(L)_SH3b2638 are identical in their N-terminal enzymatic domain. In fact, our data points to the impact of CBDs on synergy. Based on our data, the presence of very similar CBDs in the paired enzymes (such as SH3bLST and SH3bAle1) may promote synergy, whereas the combination of two different CBDs (SH3b2638 and SH3bLST) may affect it negatively. However, Schmelcher et al. have observed equally synergistic behaviour between the two chimeric PGHs λSA2-E-LysK-SH3b and λSA2-E-LST-SH3b with LST [42]. Another possible explanation is that the synergistic behaviour of the LST-derived M23 domain specifically depends on the adjacency of its native CBD, although this theory needs further exploration. It should be noted that, due to the low activity of CHAP-containing PGHs in raw milk (Appendix A), our synergy experiments were performed in UHT milk. Therefore, any potential impact of raw milk components on the observed synergistic effects cannot be detected in the chosen setting.

Intracellularly persisting *S. aureus* can be a reason for treatment failure and are associated with sub-clinical and recurring bovine mastitis [18]. Modification of LST, M23LST(L)_SH3b2638, CHAPK_SH3bLST and CHAPGH15_SH3bLST with the cell-penetrating peptide TAT significantly increased their ability to lyse *S. aureus* within infected bovine mammary gland cells (Figure 5). LST_TAT and M23_TAT were more active than CHAPK_TAT or CHAPGH15_TAT, which is in line with the other data presented in this work. The M23 domain-containing enzymes significantly reduced intracellular bacterial numbers at 250 nM (LST_TAT) and 250–500 nM (M23_TAT), whereas more than 10 times the concentration of CHAPK_TAT and CHAPGH15_TAT was needed to reach similar intracellular activity. Remarkably, LST_TAT showed stronger intracellular activity than the unmodified LST even at only one fourth of the parental concentration and even though our time-kill assay data (Appendix A) suggested that the C-terminal CPP modification leads to a reduced in vitro activity of this and other constructs. This was especially pronounced in undiluted raw milk, where LST and M23 went from complete eradication to less than a log reduction of bacterial numbers for LST_TAT and M23_TAT. This trade-off between reduced activity in milk and the enhanced ability to target intracellular bacteria could be bypassed by the application of a cocktail of modified and unmodified PGHs. This cocktail could also entail a mixture of synergistic enzymes to potentiate the treatment efficacy in bovine mastitis.

This study brought forward CHAPGH15_SH3bAle1 as a new candidate for the treatment of bovine mastitis and has further emphasised the increase in treatment efficacy that can be gained from the synergistic application of PGHs and their modification with cell-penetrating peptides.

## Figures and Tables

**Figure 1 viruses-14-02801-f001:**
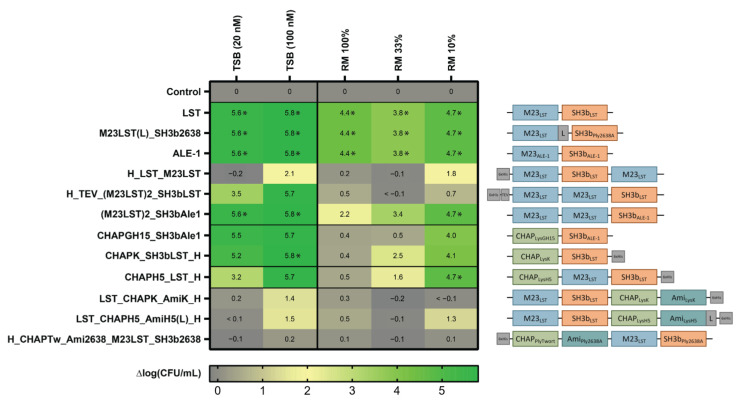
Staphylolytic activity of PGHs in growth medium and raw milk. *S. aureus* Newbould 305 (10^6^ CFU/mL) was challenged with 12 PGHs selected for high activity in raw milk in a microwell plate-based screening. PGHs were added at concentrations of 20 nM and 100 nM in TSB, and 360 nM in all raw milk conditions. Bacterial counts were determined after 180 min by plating, and average log (CFU/mL) reductions compared to the untreated control were calculated from biological triplicates. Reduction of bacterial numbers to the detection limit is indicated by (*). Black horizontal lines separate constructs according to their enzymatic activity profile (top to bottom: control, Gly-Gly endopeptidases, Gly-d-Ala endopeptidases, double mixed activity, triple mixed activity). Domain architectures of PGHs are visualized on the right (blue: M23 Gly-Gly-endopeptidase domain, green: CHAP Gly-d-Ala endopeptidase domain, teal: amidase domain, orange: SH3b-type cell wall-binding domain).

**Figure 2 viruses-14-02801-f002:**
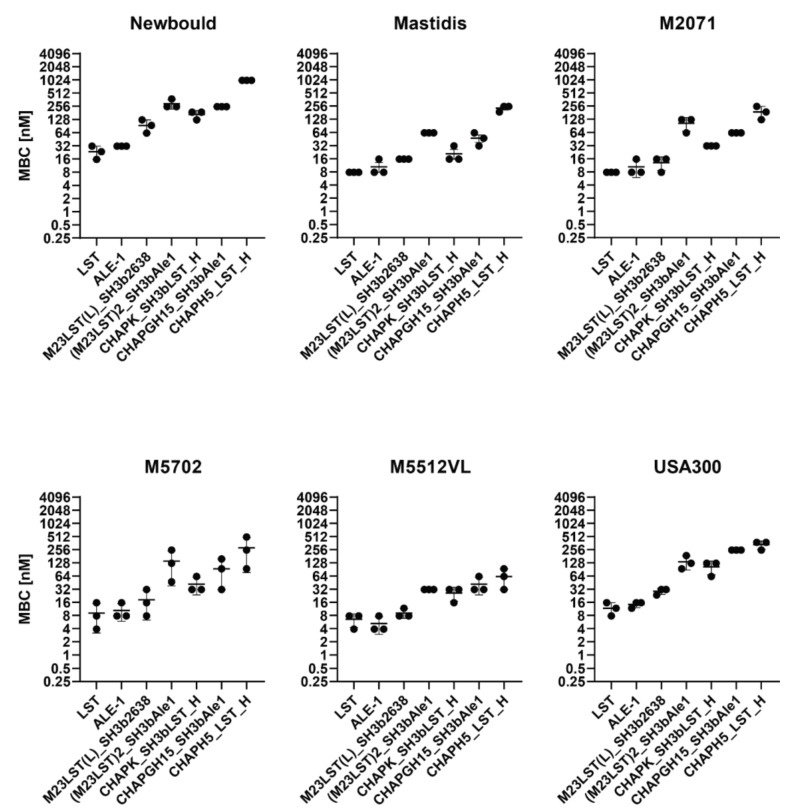
Minimum bactericidal concentrations of PGHs against six *S. aureus* mastitis isolates and the MRSA strain USA300 in UHT milk. Bacteria (2.5 × 10^5^ CFU/mL) were challenged with serially diluted PGHs in UHT milk and the MBC was determined after two hours by plating. MBCs obtained from biological triplicates (black dots) are plotted on a log2 scale (±SD).

**Figure 3 viruses-14-02801-f003:**
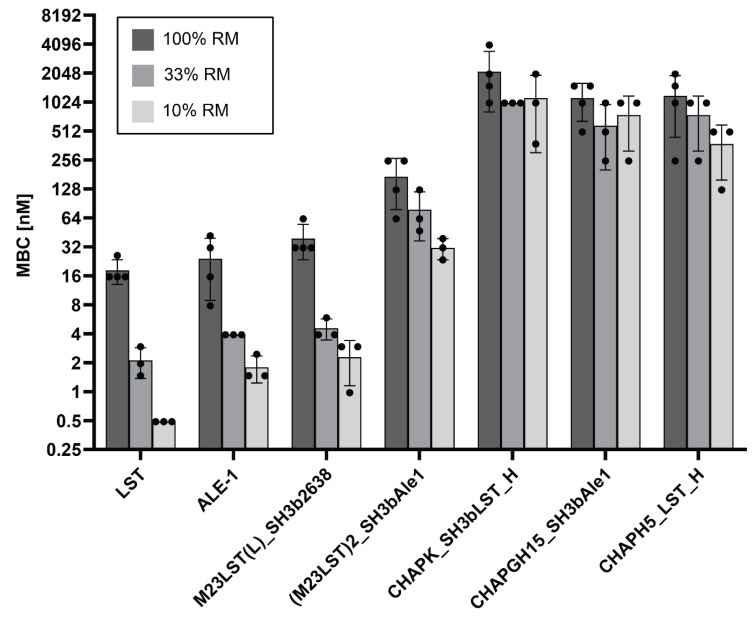
Minimum bactericidal concentrations of seven PGHs against *S. aureus* Newbould 305 in raw milk. Bacteria (2.5 × 10^5^ CFU/mL) were challenged with serially diluted PGHs in raw milk or raw milk diluted in DPBS (10% and 33%). The MBC was determined by plating after two hours. Individual values obtained from biological replicates (black dots) and bars reflecting the mean MBC are plotted on a log2 scale (±SD).

**Figure 4 viruses-14-02801-f004:**
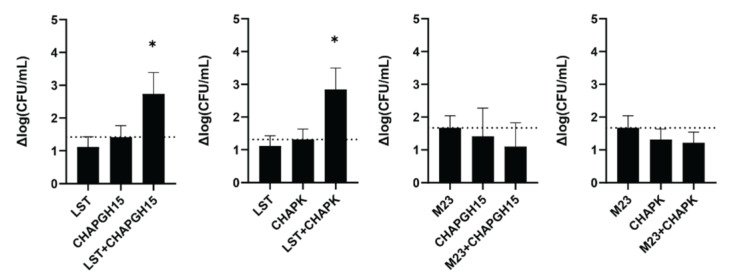
Synergistic activity of PGHs in milk. *S. aureus* Newbould 305 (10^6^ CFU/mL) was challenged with single PGHs at concentrations resulting in a similar activity in UHT milk (LST: 10 nM, M23: 10 nM, CHAPGH15: 20 nM, CHAPK: 25 nM). To show synergy, two PGHs were combined at half the concentration of the respective single treatments. Bacteria were enumerated by plating after one hour, and the average log (CFU/mL) reduction (± SEM) relative to an untreated control was calculated from six biological replicates. The dashed line indicates the activity threshold of the single enzyme with the higher activity, and mixtures with activity significantly above this line are considered synergistic. ns, *p* > 0.05; *, *p* ≤ 0.05.

**Figure 5 viruses-14-02801-f005:**
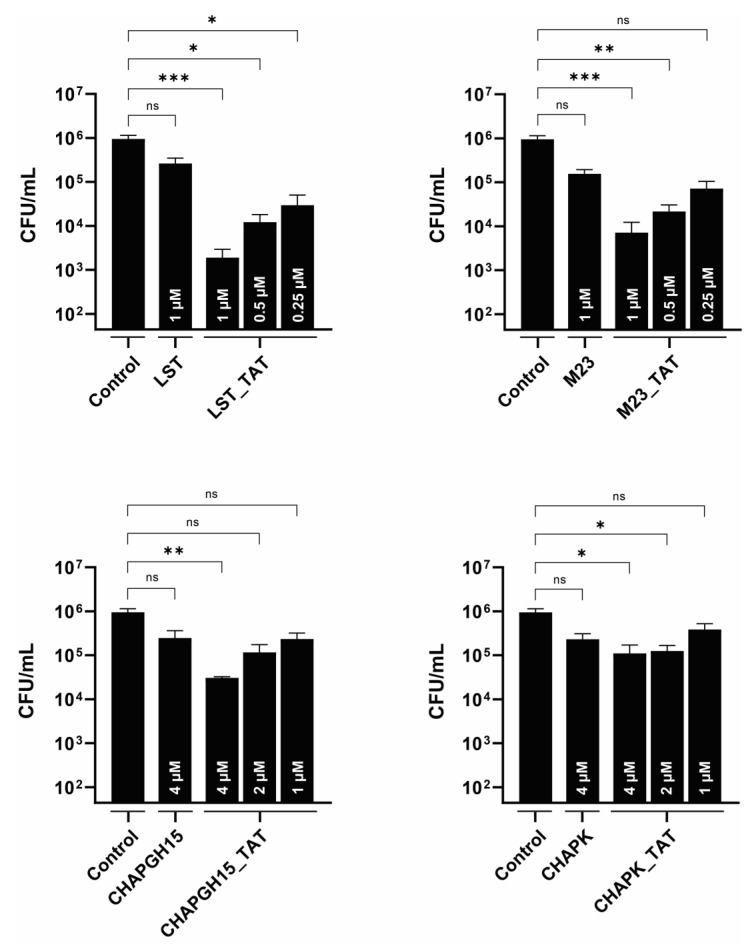
Intracellular activity of parental PGHs and PGHs modified with a cell-penetrating peptide in bovine mammary alveolar cells infected with *S. aureus* Newbould 305. MAC-T cells were infected with *S. aureus* Newbould (MOI = 50, 2 h) and treated with a parental PGH or PGH_TAT (cell-penetrating peptide) for three hours. Enzyme concentrations are indicated within each bar. Intracellular bacteria were enumerated by plating, and the average log (CFU/mL) reduction compared to an untreated control was determined (±SEM). Experiments were conducted in biological triplicates. ns, *p* > 0.05; *, *p* ≤ 0.05; **, *p* ≤ 0.01; ***, *p* ≤ 0.001.

## Data Availability

All data this manuscript can be provided by the authors upon inquiry.

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
