# Peer review of "Chimeric Peptidoglycan Hydrolases Kill Staphylococcal Mastitis Isolates in Raw Milk and within Bovine Mammary Gland Epithelial Cells"

_viruses, 2022, doi:10.3390/v14122801_

Round 1

Reviewer 1 Report

In the article “Chimeric peptidoglycan hydrolases kill staphylococcal mastitis isolates in raw milk and within bovine mammary gland epithelial cells” the authors expanded a previous work by screening novel chimeric peptidoglycan hydrolases in raw bovine milk.

The authors started by selecting the peptidoglycan hydrolases with higher staphylolytic activity in raw bovine milk, assessed their activity in vitro by time kill assays, and determined the minimum bactericidal concentration. Selected peptidoglycan hydrolases were fused to cell penetrating peptides and were shown to be promising in killing intracellular bacteria. Furthermore, the synergistic action of peptidoglycan hydrolases was tested as a way to reduce the risk of potential bacterial resistance. The study is very well designed, the methods well explained, and the results clearly presented, being of great interest for paving the way towards the use of peptidoglycan hydrolases for the treatment of bovine mastitis.

I recommend this paper for publication after a minor revision.

No language and text editing are required. My only doubt is: why did you not test the synergism in raw milk instead of UHT milk? Could not the inhibitory substances present on raw milk influence the conclusions obtained in this section?

Author Response

We thank the reviewer for the positive comments on our manuscript. Please find our answer to the question below.

Q: My only doubt is: why did you not test the synergism in raw milk instead of UHT milk? Could not the inhibitory substances present on raw milk influence the conclusions obtained in this section?

R: The reviewer is raising a valid question. Our synergy experiments were based on time-kill assays and designed in a way that the concentrations of the two PGHs to be tested (concentrations x and y) where adjusted in a way that they caused similar CFU reductions when tested individually. These effects were then compared with the CFU reduction caused by a combination of both proteins at half of these concentrations (1/2 x + 1/2 y) to test for synergy. Since the CHAP-containing PGHs included in all our synergy experiments showed virtually no activity in raw milk when tested individually (see Supplementary Figure S1), we were not able to adjust their concentrations in a way that they caused similar CFU reductions as the PGHs containing M23 domains. For this reason, we decided to conduct the synergy measurements in UHT milk, where activity of the CHAP-based constructs could be measured. This indeed presents a limitation, since possible impacts of raw milk components on the synergistic effects of the PGHs cannot be detected in this setting. The following sentence has been added to the discussion to address this limitation (L 501 after revision): “It should be noted that, due to the low activity of CHAP-containing PGHs in raw milk (Supplementary Figure S1), our synergy experiments were performed in UHT milk. Therefore, any potential impact of raw milk components on the observed synergistic effects cannot be detected in the chosen setting.”

Reviewer 2 Report

The emergence and spread of antibiotic-resistant bacteria is a serious global public health problem that affects the effectiveness of medical treatments for many infectious diseases. The situation is urgent with many bacterial pathogens, such as the ESKAPE group and others. In the present submission, the Authors set an ambitious goal to characterize a set of chimeric peptidoglycan hydrolases as effective antimicrobial agents against Staphylococcus aureus, an etiological factor of bovine mastitis. The manuscript is scientifically and technically sound and is a step forward to finding effective therapy against a potent pathogen. The experiments were performed with care for details.

Minor comments:

Introduction. The authors list lytic transglycosylases among hydrolytic enzymes (p. 2, lane 69). In fact, lytic transglycosylases exemplified by phage lambda endolysin (product of the R gene) perform non-hydrolytic cleavage of the glycan strand of bacterial murein with the formation of the non-reducing N-acetyl 1,6-anhydromuramic acid, and N-acetylglucosamine [see Eur. J. Biochem 53 (1975) 47-54; J. Bacteriol. 124 (1975) 1067-1076].

M&M. For expression vectors (p. 3, lane 114), please provide the company's name (Qiagen, …)

Results. It would be nice to provide a scheme of analyzed chimeric enzymes to ease the reader's understanding of the results delivered. Experimental data are presented in a clear systematic way.

Discussion. This chapter is really interesting and well-written.

Author Response

We thank the reviewer for the positive comments on our manuscript. Please find our answers to the minor comments below.

Q: Introduction. The authors list lytic transglycosylases among hydrolytic enzymes (p. 2, lane 69). In fact, lytic transglycosylases exemplified by phage lambda endolysin (product of the R gene) perform non-hydrolytic cleavage of the glycan strand of bacterial murein with the formation of the non-reducing N-acetyl 1,6-anhydromuramic acid, and N-acetylglucosamine [see Eur. J. Biochem 53 (1975) 47-54; J. Bacteriol. 124 (1975) 1067-1076].

R: The reviewer is absolutely right. We have removed lytic transglycosylases form this sentence, since they are not PGHs.

Q: M&M. For expression vectors (p. 3, lane 114), please provide the company's name (Qiagen, …)

R: All manufacturers’ names have now been added to the M&M section.

Q: Results. It would be nice to provide a scheme of analyzed chimeric enzymes to ease the reader's understanding of the results delivered. Experimental data are presented in a clear systematic way.

R: We think that this is an excellent idea and have added a schematic showing the domain architectures of the analyzed PGHs to Figure 1.